# Fast and Accurate Computation of the Displacement Force of Stent Grafts after Endovascular Aneurysm Repair

**DOI:** 10.3390/bioengineering9090447

**Published:** 2022-09-06

**Authors:** Ming Qing, Zhan Liu, Tinghui Zheng

**Affiliations:** 1Department of Applied Mechanics, Sichuan University, Chengdu 610065, China; 2AVIC Chengdu Aircraft Industry (Group) Co., Ltd., Chengdu 610091, China; 3Med-X Center for Informatics, Sichuan University, Chengdu 610041, China

**Keywords:** endovascular aneurysm repair, stent graft migration, displacement force, momentum theorem, computational fluid dynamics

## Abstract

*Purpose:* Currently, the displacement force of stent grafts is generally obtained using computational fluid dynamics (CFD), which requires professional CFD knowledge to perform the correct simulation. This study proposes a fast, simple, and clinician-friendly approach to calculating the patient-specific displacement force after endovascular aneurysm repair (EVAR). *Methods:* Twenty patient-specific post-EVAR computed tomography angiography images were used to reconstruct the patient-specific three-dimensional models, then the displacement forces were calculated using CFD and the proposed approaches, respectively, and their numerical differences were compared and analyzed. *Results:* Based on the derivation and simplification of the momentum theorem, the patient-specific displacement forces were obtained using the information of the patient-specific pressure, cross-sectional area, and angulation of the two stent graft ends, and the average relative error was no greater than 1.37% when compared to the displacement forces calculated by CFD. In addition, the linear regression analysis also showed good agreement between the displacement force values calculated by the new approach and CFD (R = 0.999). *Conclusions:* The proposed approach can quickly and accurately calculate the patient-specific displacement force on a stent graft and can therefore help clinicians quickly evaluate the post-EVAR displacement force.

## 1. Introduction

An abdominal aortic aneurysm (AAA) is a pathological expansion of the abdominal aortic segment, which is described as the blood vessel diameter expanding by greater than 50% or greater than 3 cm [1]. Currently, endovascular abdominal aortic aneurysm repair (EVAR) is the first preference for the therapy of AAAs [2] due to its higher early remedial impact as compared to normal open surgical treatment [3]. However, compared with normal open surgical repair, EVAR has higher risks of reintervention due to endoleaks and SG migration [4,5,6], especially in the case of complex anatomical morphology, such as short length, large-angle, and conical AAA necks [7,8,9].

The migration of SGs is greatly associated with the magnitude and direction of the displacement force (DF) on the SG [10], and if the magnitude of the DF exceeds the ability of the proximal and distal end of the SG to prevent migration, the SG may migrate, which may lead to proximal seal failure, aneurysm rupture, and limb thrombosis of the SG [11,12,13]. Therefore, the evaluation of post-EVAR DF, especially the maximum DF of the overall graft, is helpful for better stent designs and assessment of EVAR treatment outcomes [14].

Computational fluid dynamics (CFD) has been widely used in the study of AAAs to observe intravascular flow fields and to calculate the DF on the SG following EVAR [15,16,17,18]. However, although the CFD approach can obtain an accurate value of the DF by integrating the SG surface pressure and wall shear stress [19,20,21], its procedure is complicated and tedious, as follows: (1) First, the computational model needs to be reconstructed based on patient-specific CTA images. (2) Second, the 3D models need to be meshed appropriately. (3) Third, time-consuming calculations and simulations with appropriate boundary conditions must be carried out to obtain the pressure field and wall shear stress on the SG. (4) Finally, the pressure and wall shear stress are integrated to obtain the DF. For these procedures, whether 3D model meshing, calculation, or post-processing, professional CFD background knowledge is required, but clinicians generally lack knowledge of CFD techniques. Therefore, a fast, simple, and clinician-friendly approach to obtaining the patient-specific DF on the SG is urgently needed for clinicians to judge whether the SG may migrate after the operation.

Assuming blood is steady and non-viscous, Liffman et al. and Mohan et al. developed a mathematical model for studying the DFs on a planer and symmetrical bifurcated graft using continuity and momentum equations [22,23]. However, due to their simplifying assumptions about the blood and graft geometry, their theoretical model can only give a rough estimation of the axial force acting on a graft; it seems feasible to use the momentum theorem to directly solve the DF. Encouraged by their preliminary studies and aimed at proposing a fast, simple, and clinician-friendly approach obtaining the DF on the patient-specific SG model, the present study first starts from the fundamental theory of fluid mechanisms to derive a reasonable simplified formula based on the momentum theorem, then twenty patient-specific post-EVAR computed tomography angiography (CTA) images were used to reconstruct the patient-specific three-dimensional (3D) models. Finally, the DFs were calculated using CFD and the proposed approach, respectively, and their numerical differences were compared and analyzed. 

## 2. Materials and Methods

This was a retrospective observational study that was conducted following the principles of the Helsinki Declaration and met the requirements of medical ethics. The Ethical Review Committee of the West China Hospital of Sichuan University (Chengdu, Sichuan, China) approved this research. Since all the data were collected from retrospective anonymized databanks and as our study was purely observational, patient approval and informed consent were waived.

### 2.1. Geometry

Twenty patient-specific SG models, shown in Figure 1, were established using thin-slice CTA images (scanner: SIEMENS/SOMATOM Definition Flash, slice thickness: 1.0 mm, pixel size 0.6133 mm) with the commercially available Mimics software (version 15.0; Materialise, Plymouth, UK).

### 2.2. CFD Computation

#### 2.2.1. Governing Equations 

The blood was assumed to be incompressible, laminar, homogenous, and Newtonian [24,25], with the corresponding governing equations given as:(1)$$\rho \frac{d\vec{u}}{dt}=-\nabla p+\mu \Delta \vec{u}$$
(2)$$\nabla \cdot \vec{u}=0$$
where $\vec{u}$ and p represent, respectively, the fluid velocity vector and the pressure. ρ and μ are the density 1050 kg/m^3^ and dynamic viscosity 3.5 × 10^−3^ kg m/s of the blood, respectively [26,27].

#### 2.2.2. Boundary Conditions 

The no-slip condition was imposed on the surface of the models according to the rigid wall assumption [28]. It was shown that the maximum DF and peak pressure drop arose at peak pressure throughout the cardiac cycle [29,30]. As a result, a constant velocity of v = 0.1 m/s was applied at the inlets, and the pressure was set as 105 mmHg at the outlets, which characterized the moment of peak pressure throughout the cardiac cycle [31].

#### 2.2.3. Meshing and Computing

Computational meshes were established using ANSYS ICEM 14.5 (ANSYS, Inc., Canonsburg, Pennsylvania, PA, USA). The meshes contained a mixture of tetrahedral and hexahedral volume meshes. As a result of the boundary layer effect of fluid, the mesh near the wall needed to be densified. To decrease the difference in the calculation results caused by the different degrees of mesh refinement, we adopted the Grid Independence Index (GCI) to evaluate the generated mesh [32]. When the GCI of all the test variables was less than 2%, it was considered that there was a small enough spatial discretization error without further mesh refinement [33,34]. The ultimate number of cells for the meshes ranged from 1.5 million to 2.0 million.

The CFD calculations were performed using ANSYS Fluent 14.5 (ANSYS, Canonsburg, Sylvania, Pennsylvania, PA, USA). The calculation model was a laminar flow model. A simple algorithm was used as the calculation method for pressure and velocity coupling; the second-order upwind scheme was used to discretize the momentum equation and energy equation; the finite volume method was used to solve the continuity equation and momentum equation.

#### 2.2.4. Calculation of Displacement Force 

The DF was calculated by taking an area integral of the net pressure and wall shear stress (WSS) over the entire wall of the SG:(3)$$\mathrm{Displacement}\;\mathrm{force}:\;F=\int_{A}^{\;}pdA+\int_{A}^{\;}\vec{\tau }dA\;$$
where p is pressure, $\vec{\tau }$ is the WSS vector, $\int_{A}^{\;}pdA$ is the pressure force, and $\int_{A}^{\;}\vec{\tau }dA$ is the WSS force.

The angle between the DF vector and the positive direction of the space rectangular coordinate system can be calculated by the following formula:(4)$$\theta_{x}=arccos\left(\frac{F_{x}}{\sqrt{F_{x}^{2}+F_{y}^{2}+F_{z}^{2}}}\right)\times \frac{180}{\pi }\;$$
(5)$$\theta_{y}=arccos\left(\frac{F_{y}}{\sqrt{F_{x}^{2}+F_{y}^{2}+F_{z}^{2}}}\right)\times \frac{180}{\pi }$$
(6)$$\theta_{z}=arccos\left(\frac{F_{z}}{\sqrt{F_{x}^{2}+F_{y}^{2}+F_{z}^{2}}}\right)\times \frac{180}{\pi \;}$$
where $F_{x}$*,*
$F_{y}$*,* and $F_{z}$ are the component sizes of the DF projected on the x-, y-, and z-axes respectively. $\theta_{x}$, $\theta_{y}$, and $\theta_{z}$ are the angles between the DF vector and the positive direction of the x-, y-, and z-axes, respectively. The x-direction is the right (positive) and left (negative) direction; the y-direction is the front (positive) and rear (negative) direction; the z-direction is the up (positive) and down (negative) direction.

### 2.3. The Proposed Approach to Computing DF 

#### 2.3.1. Governing Equations

The DF can be also calculated by the momentum theorem, which only needs the flow information at the inlet and outlet when the flow is steady. Thus, we may calculate the DF using the momentum theorem as follows:(7)$$\vec{F}=\left(p_{1}A_{1}+\rho A_{1}{\vec{V}}_{1}^{2}\right)\vec{n_{1}}+\left(p_{2}A_{2}-\rho A_{2}{\vec{V}}_{2}^{2}\right)\vec{n_{2}}+\left(p_{3}A_{3}-\rho A_{3}{\vec{V}}_{3}^{2}\right)\vec{n_{3}}$$
where $p_{i}$, $A_{i}$, $\vec{V_{i}}$, and $\vec{n_{i}}$ (*i* = 1, 2, 3) represent the pressure, area, blood flow velocity vector, and unit normal vector of the inlet and outlet sections, respectively, as shown in Figure 2a. The length of the unit normal vector is 1, which mainly represents the spatial angle of the section, so the unit normal vector can also be expressed as
(8)$$\vec{n_{i}}=\left(cos\alpha_{i},\;cos\beta_{i},cos\gamma_{i}\right)$$
where α, β, and γ are the angles between the normal vector and the coordinate axis, as shown in Figure 2b.

In this study, the direction of the unit normal vector was always perpendicular to the inlet and outlet section towards the SG.

If we substitute the boundary condition of CFD into Equation (7), the ratio $p_{i}A_{i}/\rho A_{i}U_{i}^{2}$ ranges from 76 to 16,396, indicating that the DF caused by the momentum change is negligibly small as compared to the pressure force. Therefore, Equation (7) can be simplified as follows:(9)$$\vec{F}=\left(p_{1}A_{1}\right)\vec{n_{1}}+\left(p_{2}A_{2}\right)\vec{n_{2}}+\left(p_{3}A_{3}\right)\vec{n_{3}}$$

According to Equation (9), the DF of the SG is decided by the pressure force difference between the two SG ends (Figure 2c). However, it was found that the DF contributed by the pressure drop (ranging from 0.25 to 2.86 mmHg in the current models) only accounted for less than 2% of the total DF. Therefore, Equation (9) may be further simplified to Equation (10).
(10)$$\vec{F}=p\left(A_{1}\vec{n_{1}}+A_{2}\vec{n_{2}}+A_{3}\vec{n_{3}}\right)$$
where p is the patient-specific systolic blood pressure. As a result, as long as the pressure, the cross-sectional area, and the angle of the SG ends are measured, the patient-specific DF can be quickly obtained. 

#### 2.3.2. Measurement of the Cross-Sectional Area and Angles of SG Ends

(1) Measurement of the cross-sectional area of the SG ends: The model’s surface area was derived directly from the Mimics software. Therefore, after the SG model (**model 1**) was established, any part (**model 2**) could be extended from the section to obtain the whole of **model 3**. Using the Boolean operation between the three models, the cross-sectional area was obtained (Figure 3a).
Cross-sectional area = (model 1 surface area + model 2 surface area − model 3 surface area)/2. (11)

(2) Measurement of the normal vector of the section: We determined the coordinates of three points on the cross-section at both ends of the SG model, and then, the start vector and end vector were determined according to the three points (Figure 3b).
(12)$$\vec{\mathrm{Normal}\;\mathrm{vector}}=\vec{\mathrm{Start}\;\mathrm{vector}}\times \vec{\mathrm{Termination}\;\mathrm{vector}}.$$

### 2.4. Statistical Analysis

The SPSS software (v21.0, IBM Inc., New York, NY, USA) was used for the statistical analysis. Comparing the relative error (relative error = absolute error/true value × 100%) between the total DF calculated by the momentum theorem and the CFD method, it was considered that the value calculated by the CFD method was the true value in this study. Linear regression analysis was carried out to evaluate the correlation between the simplified formula of the momentum theorem and the CFD method. The coefficient of determination (R-value) was used to measure the fit of the predicted value to the true value. The closer the R-value was to 1, the closer the predicted value was to the true value. A Bland–Altman analysis was carried out to compare the two measurement methods; additionally, the average deviation and limits of agreements (defined as 1.96× standard deviations) are provided [35,36].

## 3. Results

The magnitude of the DF calculated using the momentum theorem (Equation (10)) and CFD method (Equation (3)) is shown in Table 1, and the angle of the DF calculated is shown in Table 2. The average relative error of the DF received by the momentum theorem and CFD method was 1.37%. In particular, the maximum relative error in the model of SG6 was 6.91%. On $\theta_{x}$ and $\theta_{z}$, the average relative errors of all the models were 1.04% and 1.43% respectively, whilst the maximum relative errors in the model of SG6 were 13.60% and 19.91%, respectively. The relative error of all the models on $\theta_{y}$ was small; the average relative error was 0.21%, and the maximum relative error was only 1.04%.

A linear regression analysis was carried out on every calculated DF dataset to further evaluate the correlation between the two methods. Figure 4 suggests that there was a precise consistency between the magnitude of the DF calculated by the simplified momentum theorem and the CFD method (R = 0.999), as well as the DF angle (R > 0.993).

The Bland–Altman plot (Figure 5) shows that there was only a small deviation between the magnitude of the DF calculated by the momentum theorem and the CFD method (the upper and lower limits of DF consistency calculated by the two methods were 0.111 N and −0.13 N, respectively); only the DF of SG6 exceeded the limit. The distinction between the DF angles calculated by the two methods was additionally very small (the upper and lower limits of $\theta_{x}$ consistency were 5.71° and −7.19°, respectively; the upper and lower limits of $\theta_{y}$ consistency were 0.68° and −0.83, respectively; the upper and lower limits of $\theta_{z}$ consistency were 5.97° and −8.05, respectively). The SG6 patient exceeded the limits of $\theta_{x}$ and $\theta_{z}$. The SG7 and SG8 patients slightly exceeded the limit on $\theta_{y}$.

## 4. Discussions

Endovascular abdominal aortic aneurysm repair (EVAR) is used to insert an interventional device with a stent graft (SG) from the femoral artery and then delivering it to the subrenal position to release the stent. In this way, a new flow channel is constructed to isolate the blood flow, prevent the blood from directly scouring the aneurysm wall, and treat an abdominal aortic aneurysm. Since Volodos et al. first established EVAR technology 1987 [37], it has since become more mature with continuous improvements over more than 30 years to the structure and anchoring method of the SG [38,39,40,41]. However, the use of EVAR is prone to causing endoleaks and stent migration when AAA patients have unfavorable anatomies such as severe aneurysm necks.

At present, CFD is used to obtain the DF on the SG, the acting force produced by the blood to the SG, which is the main cause for the SG’s migration. However, the CFD-based DF calculation method requires professional CFD knowledge for accurate simulation. Moreover, the conventional CFD approach is complex and time consuming [42]. In this study, we proposed a simple and fast approach to obtaining the global DF using the momentum equation, which only needs the patient-specific blood pressure, the cross-sectional areas, and the angulations of the proximal and distal ends of the SG. The results showed that the DFs calculated by the proposed simplified formula were in good agreement with those obtained by CFD (R > 0.993).

Agreeing with previous CFD studies [23,29,42], the current study showed that the contribution of the change in blood flow velocity to DF is negligibly small. Therefore, the blood momentum inside the SG can be regarded as unchanged, and the DF of an SG is mainly decided by the pressure force difference at its two ends (Equation (9)). This may explain why the DF waveform trend of SGs is always consistent with that of the pressure waveform in previous CFD research [43,44,45]. In addition, although the force is a vector and the change in its magnitude and direction both may contribute to the force difference, this study indicated that the contribution by the pressure drop between the proximal and distal SG accounted for less than 2% of the total DF. That is, the DF on the SG is mainly due to the direction change of pressure forces at the SG ends, and it is reasonable to evaluate the DF using the patient-specific blood pressure (Equation (10)). Furthermore, because the pressure force is the product of the pressure and area, high blood pressure and large areas of SG anchor ends may both cause a large pressure force and, thus, a potentially large DF. As a result, not only is hypertension a risk factor for SG migration after EVAR [22,29,42], but an increase in the area of the proximal and distal SG can also result in an increased risk of adverse events [22,46]. In practice, a proximal oversizing ratio of greater than 30% has been suggested to lead to the risk of stent migration [47,48], and the use of a clock bottom design at the distal end was found to increase the incidence of type Ib endoleaks [49]. 

This study revealed that the angulation of the proximal and distal SG plays a key role in the total DF on the SG. If the angulations of the proximal and distal SG are not significantly different, the pressure forces at the two ends will cancel each other out, resulting in a small DF on the SG (the SG8, SG9, and SG19 patients; Table 1). On the contrary, if the angulation of the two anchor ends is very different, the DF on the SG will be large (the SG4 and SG18 patients; Table 1). Therefore, minimizing the angle difference between the two SG ends can reduce the risk of displacement following EVAR. Previous studies have indicated that changes in the aorta angulation caused changes in the magnitude and direction of the displacement force [19,50,51]. This may be because the current SG placement generally allows the anchor end to conform to the shape of the blood vessels, so that the DF of the SG will be directly affected by the aortic morphology. 

There are still some deficiencies in our research. First of all, there were only 20 patient-specific models reconstructed and tested. Although the results reflect the accuracy of our measurements and calculations to some extent, they need to be verified by a much larger cohort of patients. Secondly, because the CFD calculation showed that the pressure drops of the two SG ends were small, the proposed formula did not account for the contribution of the pressure differences. However, if an implanted SG is excessively distorted, the pressure drop between the proximal and distal ends will increase, and its contribution to the DF will increase. Additionally, the current study showed that if the pressure difference increases by 1mmHg, the calculation error by the simplified equation will increase by 0.8%. Therefore, in the case of the high distortion of an SG, the displacement force caused by a pressure drop should be properly considered. Last but not least, the density and dynamic viscosity used in the CFD simulations represent parameters of cold blood (at a temperature of about 4 ℃). There may be some errors if the parameters of living humans are not used. However, the simplified formula removes the term regarding density. In addition, the human blood viscosity ranges from 0.0040~0.0050 kg·m/s, so such an error should be relatively small and within a controllable range.

## 5. Conclusions

In this study, we proposed a simplified formula to calculate the DF of the SG. This approach only requires the section parameters of the 3D-reconstructed SG model, and then, the DF of the SG can be directly obtained by a simple calculation. Compared with the traditional CFD method, this new approach achieved a similar level of accuracy without the need for complex model post-processing or computer simulations. 

## Figures and Tables

**Figure 1 bioengineering-09-00447-f001:**
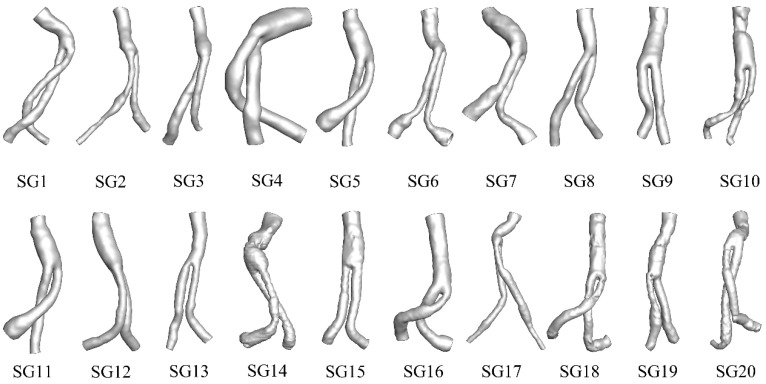
Schematic diagram of the patient-specific models.

**Figure 2 bioengineering-09-00447-f002:**
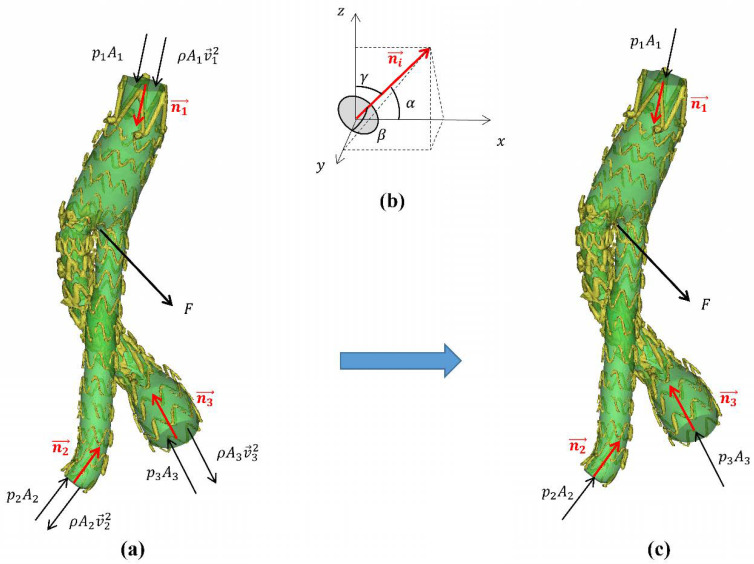
(**a**) Schematic diagram of each part of the momentum theorem formula. The red arrow indicates the direction of the unit normal vector; the black arrow indicates the direction of the force; p represents the pressure; $\rho A{\vec{V}}^{2}$ represents momentum; $\vec{n_{i}}$ represents the unit normal vector; F represents the displacement force. (**b**) Schematic diagram of the unit normal vector. The normal vector represents the direction of the plane and is defined as the cosine value of the angle between the positive direction of the coordinate axis and the vector. (**c**) Simplified stress diagram of a stent-graft; the displacement force is only related to the pressure, area, and angle of the inlet and outlet section.

**Figure 3 bioengineering-09-00447-f003:**
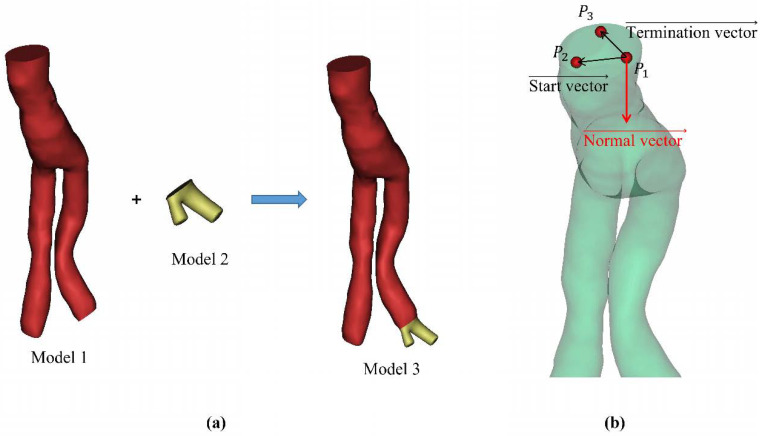
(**a**) Schematic diagram of section area measurement method. (**b**) Schematic diagram of the angle measurement method. The normal direction was always toward the inside of the stent graft.

**Figure 4 bioengineering-09-00447-f004:**
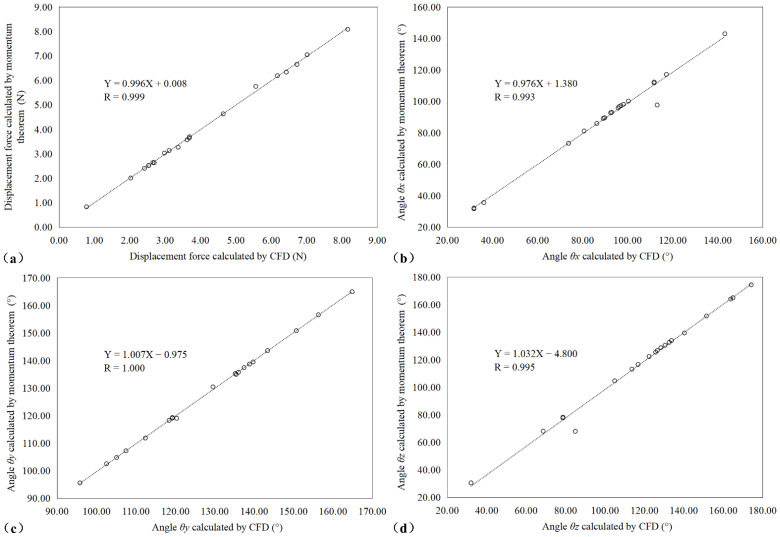
Linear scatter diagram showing the correlation between the magnitude of (**a**) displacement force (**b**) $\theta_{x}$, (**c**) $\theta_{y}$, and (**d**) $\theta_{z}$ as calculated by the simplified momentum theorem and CFD method.

**Figure 5 bioengineering-09-00447-f005:**
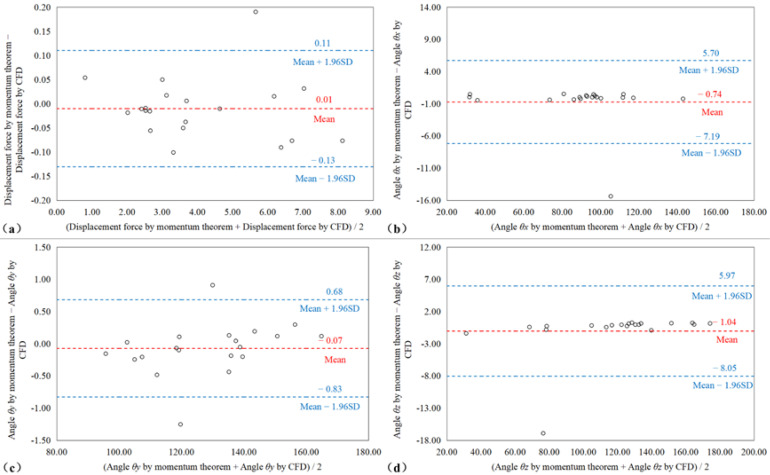
Bland–Altman diagram comparing the data consistency of (**a**) the displacement force, (**b**) $\theta_{x}$, (**c**) $\theta_{y}$, and (**d**) $\theta_{z}$ as calculated by the simplified momentum theorem and CFD method.

**Table 1 bioengineering-09-00447-t001:** The displacement force as calculated by the CFD method and simplified momentum quantitative theorem.

Heading	CFD Method (N)				Simplified Momentum Quantitative (N)				Relative Error (%)
	*F_x_*	*F_y_*	*F_z_*	*F*	*F_x_*	*F_y_*	*F_z_*	*F*	
SG1	4.49	−2.12	−2.51	5.56	4.67	−2.15	−2.58	5.75	3.42
SG2	–0.37	−2.39	−1.74	2.98	−0.38	−2.44	−1.76	3.03	1.67
SG3	0.87	−2.72	–1.25	3.11	0.9	−2.73	−1.24	3.13	0.55
SG4	−5.62	−3.34	2.56	7.02	−5.62	−3.35	2.62	7.05	0.45
SG5	3.14	−1.80	0.73	3.69	3.12	−1.81	0.78	3.7	0.16
SG6	−0.31	−0.71	0.07	0.78	−0.11	−0.76	0.31	0.83	6.91
SG7	0.59	−1.83	3.07	3.62	0.55	−1.74	3.07	3.57	1.39
SG8	−0.13	−1.61	−1.95	2.53	−0.14	−1.64	−1.92	2.53	0.36
SG9	0.01	−0.24	−2.40	2.42	0.02	−0.24	−2.39	2.41	0.45
SG10	−0.48	−0.73	−3.25	3.37	−0.46	−0.71	−3.16	3.27	3
SG11	3.13	−1.80	0.72	3.68	3.1	−1.78	0.73	3.65	1.02
SG12	0.16	−1.83	−1.75	2.54	0.18	−1.81	−1.75	2.52	0.56
SG13	−0.20	−0.53	−1.95	2.03	−0.20	−0.52	−1.93	2.01	0.92
SG14	−0.27	−5.96	−1.59	6.18	−0.30	−5.98	−1.58	6.19	0.25
SG15	−0.73	−5.07	−4.37	6.73	−0.77	−5.00	−4.31	6.65	1.14
SG16	−2.12	−3.30	−2.49	4.64	−2.11	−3.30	−2.48	4.63	0.23
SG17	−1.00	−0.81	−2.37	2.69	−0.98	−0.78	−2.32	2.64	2.08
SG18	−3.03	−5.82	−4.86	8.17	−3.06	−5.72	−4.83	8.09	0.94
SG19	−0.48	−2.03	−1.65	2.66	−0.47	−2.01	−1.65	2.64	0.59
SG20	0.09	−4.73	−4.35	6.43	0.09	−4.67	−4.28	6.34	1.41

*F* is the displacement force; *F_x_*, *F_y_* and *F_z_* represent the components of *F* in the x-, y-, and z-directions, respectively. Relative error (%) = absolute value of (CFD value − momentum value)/CFD value × 100%.

**Table 2 bioengineering-09-00447-t002:** The CFD method and simplified momentum quantitative used to calculate the displacement force angle.

Heading	CFD Method (°)			Simplified Momentum Quantitative (°)			Relative Error (%)		
	*θx*	*θy*	*θz*	*θx*	*θy*	*θz*			
SG1	35.71	111.92	116.66	36.18	112.41	116.79	1.29	0.43	0.11
SG2	97.26	143.59	125.45	97.04	143.4	125.7	0.22	0.13	0.2
SG3	73.37	150.82	113.25	73.77	150.7	113.68	0.54	0.08	0.38
SG4	142.94	118.35	68.21	143.17	118.42	68.62	0.16	0.06	0.6
SG5	32.28	119.37	77.82	31.81	119.27	78.64	1.46	0.09	1.04
SG6	97.72	156.59	68.04	113.1	156.29	84.96	13.6	0.19	19.91
SG7	81.18	119.09	30.65	80.67	120.35	32.05	0.64	1.04	4.36
SG8	93.08	130.47	139.36	92.96	129.56	140.28	0.13	0.7	0.66
SG9	89.57	95.65	174.34	89.81	95.8	174.2	0.26	0.16	0.08
SG10	98.15	102.58	164.94	98.15	102.56	164.96	0	0.02	0.01
SG11	31.78	119.14	78.42	31.76	119.24	78.68	0.07	0.08	0.34
SG12	85.98	135.83	133.89	86.33	136.01	133.75	0.4	0.14	0.1
SG13	95.77	104.86	164.01	95.77	105.1	163.78	0.01	0.23	0.14
SG14	92.74	164.98	104.76	92.46	164.86	104.93	0.3	0.07	0.16
SG15	96.69	138.78	130.43	96.25	138.83	130.48	0.46	0.04	0.04
SG16	117.11	135.35	122.34	117.21	135.22	122.4	0.08	0.1	0.05
SG17	111.7	107.28	151.68	111.75	107.49	151.5	0.04	0.19	0.12
SG18	112.26	135.01	126.65	111.78	135.45	126.54	0.42	0.32	0.09
SG19	100.19	139.54	128.63	100.35	139.74	128.37	0.16	0.15	0.2
SG20	89.18	137.48	132.51	89.19	137.44	132.55	0	0.03	0.03

*θx*, *θy*, and *θz* represent the angles between the force vector *F* and the positive direction of the rectangular coordinate system. Relative error (%) = absolute value of (CFD value − momentum value)/CFD value × 100%.

## Data Availability

The original contributions presented in the study are included in the article, further inquiries can be directed to the corresponding author.

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
