# Peer review of "Fast and Accurate Computation of the Displacement Force of Stent Grafts after Endovascular Aneurysm Repair"

_bioengineering, 2022, doi:10.3390/bioengineering9090447_

Round 1
Reviewer 1 Report
1. Abbreviations should not be used in the Abstract, unless absolutely necessary
2. Some parts of the text are written in very poor English that is difficult to understand. This particularly concerns Introduction and Discussion. The manuscript needs extensive proofreading. Considering the fact that The Authors have probably no experience with medical texts, they should ask for proofreading a colleague with basic knowledge of endovascular procedures for abdominal aortic aneurysms.
3. Page 3, line 88. Density and dynamic viscosity used in simulations represent such parameters of cold blood (at temperature about 4O C). The Authors should either use this parameters found in living humans, or at least should discuss it as a drawback of the study
4. Page 9, line 225. Actually first implantations of aortic stentgrafts were done by Ukrainian vascular surgeon Nicolai L. Volodos in 1987
Author Response
1) Abbreviations should not be used in the Abstract, unless absolutely necessary.
Response:
Thank you for your valuable advice. We have removed unnecessary abbreviations from the abstract according to your requirements.
2) Some parts of the text are written in very poor English that is difficult to understand. This particularly concerns Introduction and Discussion. The manuscript needs extensive proofreading. Considering the fact that The Authors have probably no experience with medical texts, they should ask for proofreading a colleague with basic knowledge of endovascular procedures for abdominal aortic aneurysms..
Response:
Thank you very much for your suggestion, we carefully checked the revised manuscript and resorted to The Charlesworth Group for the language-editing service.
3) Page 3, line 88. Density and dynamic viscosity used in simulations represent such parameters of cold blood (at temperature about 4℃). The Authors should either use this parameters found in living humans, or at least should discuss it as a drawback of the study.
Response:
Thank you for your constructive comments. According to your request, we added this point in the discussion on line 278 on page 10: "Last but no the least, density and dynamic viscosity used in CFD simulations represent such parameters of cold blood (at temperature about 4℃). There may be some errors if the parameters of the living humans are not used. However, since the simplified formula removes the term about density, such error should be relatively small and within a controllable range."
4) Page 9, line 225. Actually first implantations of aortic stentgrafts were done by Ukrainian vascular surgeon Nicolai L. Volodos in 1987.
Response:
Thank you for your valuable advice.This is our mistake. We have changed line 25 on page 9 to "Since Volodos et al. first started the technology of EVAR in 1987(37), the EVAR technology is turning into greater maturity with the continuous improvement of the structure and anchoring method of the SG for extra than 30 years."
References
- Volodos N, Karpovich IP, Shekhanin VE, Troian V, Iakovenko LF. A case of distant transfemoral endoprosthesis of the thoracic artery using a self-fixing synthetic prosthesis in traumatic aneurysm. Grudnaia khirurgiia (Moscow, Russia). 1988;6:84-6.

Reviewer 2 Report
Very interesting study and conclusions about the subject
Author Response
Thank you very much for your review of this article.

Reviewer 3 Report
This is an interesting study where the authors introduced their simplified method to calculate the DF of stent grafts versus the commonly used CFD method. Using post-EVAR CTA images from twenty patients, the authors compared their method versus the CFD method in computing the DF. They argued that their method is superior to the CFD method, where the DFs calculated are comparable between these two methods. Overall, this is a simple study to introduce a new, simplified approach to calculate the DF versus the commonly used CFD method. This study will provide a stepping stone to clinicians who would like to try this method instead of the CFD method.
Author Response

(The authors gave the same response as above.)
